# Plastic Responses of *Iris pumila* Functional and Mechanistic Leaf Traits to Experimental Warming

**DOI:** 10.3390/plants14060960

**Published:** 2025-03-19

**Authors:** Katarina Hočevar, Ana Vuleta, Sanja Manitašević Jovanović

**Affiliations:** Department of Evolutionary Biology, Institute for Biological Research “Siniša Stanković”—National Institute of the Republic of Serbia, University of Belgrade, 11108 Belgrade, Serbia; katarina.hocevar@ibiss.bg.ac.rs (K.H.); ana.vuleta@ibiss.bg.ac.rs (A.V.)

**Keywords:** experimental warming, thermal acclimation, phenotypic plasticity, functional leaf traits, mechanistic leaf traits, *Iris pumila* L.

## Abstract

Phenotypic plasticity is an important adaptive strategy that enables plants to respond to environmental changes, particularly temperature fluctuations associated with global warming. In this study, the phenotypic plasticity of *Iris pumila* leaf traits in response to an elevated temperature (by 1 °C) was investigated under controlled experimental conditions. In particular, we investigated important functional and mechanistic leaf traits: specific leaf area (SLA), leaf dry matter content (LDMC), specific leaf water content (SLWC), stomatal density (SD), leaf thickness (LT), and chlorophyll content. The results revealed that an elevated temperature induced trait-specific plastic responses, with mechanistic traits exhibiting greater plasticity than functional traits, reflecting their role in short-term acclimation. SLA and SD increased at higher temperatures, promoting photosynthesis and gas exchange, while reductions in SLWC, LDMC, LT, and chlorophyll content suggest a trade-off in favor of growth and metabolic activity over structural investment. Notably, chlorophyll content exhibited the highest plasticity, emphasizing its crucial role in modulating photosynthetic efficiency under thermal stress. Correlation analyses revealed strong phenotypic integration between leaf traits, with distinct trait relationships emerging under different temperature conditions. These findings suggest that *I. pumila* employs both rapid physiological adjustments and longer-term structural strategies to cope with thermal stress, with mechanistic traits facilitating rapid adjustments and functional traits maintaining ecological stability.

## 1. Introduction

Human activity, primarily the production of greenhouse gas emissions, has undeniably driven global warming, with global surface temperatures rising 1.1 °C above pre-industrial levels between 2011 and 2020 [1]. Rising temperatures create new climatic conditions that affect the functioning and distribution of species. Human-induced climate change has already led to numerous negative impacts on natural ecosystems, including disruption of key ecological processes, changes in the functioning of ecosystems, and a decline in biodiversity [2,3,4]. As the climate changes, plant populations may no longer be optimally adjusted to new conditions and their survival depends on their capacity to adequately respond to altered environments [4]. If global warming continues, numerous species could face extinction, as the environments to which they have adapted over the course of time will change within a few decades [5]. Therefore, understanding plant responses to rapid climate change is essential for comprehending both the changes that have already occurred and those likely to take place in the near future.

To persist in a rapidly changing climate, plant organisms may exploit different coping mechanisms. They can move to more favorable habitats through seed dispersal, adapt through natural selection, or adjust to new conditions through phenotypic plasticity [6]. If the rate of environmental change is greater than the rate of evolutionary response, phenotypic plasticity is a likely mechanism for such a circumstance [7,8,9]. If traits are plastic, and if this plastic change shifts trait values in a direction that maintains fitness, this may be sufficient to anticipate or prevent extinction [9]. To date, most empirical evidence suggests that phenotypic plasticity is the predominant mechanism by which natural plant populations cope with rapid climate warming within their distribution range [10,11,12]. Phenotypic plasticity is considered to facilitate the survival of plant populations in novel environments, enhancing their ecological success and adaptability by enabling the development of phenotypes better suited to changing conditions [13,14,15,16]. Plastic responses in plants can have significant implications for various aspects of their life history, including reproductive success, dispersal patterns, and interactions between species [17].

Among plant traits, leaf traits are particularly important in associating plant resource use and biomass production with ecosystem functioning [18,19] and are, therefore, expected to be especially affected by rising temperatures due to global warming [20,21,22]. Functional leaf traits encompass morpho-physio-phenological characteristics that directly influence an individual’s growth, reproduction, and survival by shaping its interactions with the environment. In contrast, mechanistic leaf traits are those with clearly defined physiological roles, providing a more specific understanding of processes like resource uptake and utilization [23]. Examples of functional traits include leaf area and specific leaf weight, while mechanistic traits encompass photosynthetic capacity and transpiration rates. This classification offers a comprehensive framework for investigating plant adaptive strategies under varying environmental conditions, which is essential for understanding ecological responses to climate warming.

Specific leaf area (SLA) is the major determinant of a plant’s ecological function, since it reflects the expected return of previously captured environmental resources [18,24,25]. It is highly responsive to environmental factors, such as irradiance [26,27], water availability [28], atmospheric carbon dioxide [29], ozone concentration [30], soil fertility, and nutrient supply [31,32,33], while the relationship between SLA and temperature has not yet been sufficiently explored [34]. Leaf dry matter content (LDMC), another important functional trait, is related to leaf protein content and cytoplasmic volume, which reflect the structure of leaf cells [35]. LDMC is positively related to leaf longevity and negatively related to potential relative growth rate [19]. The relationship between LDMC and temperature is complex and varies considerably across different species and ecosystems [36,37,38]. Specific leaf water content (SLWC), a mechanistic trait most consistently correlated with climate, exhibits a significant relationship with leaf tissue density and leaf thickness [39]. SLWC is strongly influenced by temperature and generally exhibits a positive correlation with it under optimal environmental conditions [40]. Leaf thickness (LT) is a mechanistic leaf trait that is also related to a species’ resource acquisition and utilization strategies and affects both light absorption and carbon dioxide diffusion rate [41,42]. Negative relationships have been observed between LT and both photosynthetic rate [43,44] and growth rate [45,46], making LT a useful screening tool for assessing a species’ productivity [47] or overall ecological performance [48]. Under the influence of global warming, LT is expected to decrease as higher temperatures cause increased transpiration rates and reduced cell expansion, resulting in thinner leaves [49]. Stomatal density (SD), a mechanistic leaf trait, is widely recognized as an indicator of a plant’s gas exchange and transpiration potential [50,51,52,53,54]. Environmental factors, especially temperature, play a crucial role in influencing stomatal density. Higher temperatures can lead to an increase in SD, which can contribute to increased transpiration and leaf cooling as plants adjust to the changing conditions. Nevertheless, it is important to acknowledge that stomatal density alone is not sufficient to draw definitive conclusions about transpiration and stomatal regulation, as the dynamic responsiveness of stomata to environmental factors often plays a more decisive role [55,56]. Chlorophyll content plays a central role in photosynthesis and serves as an important physiological marker for assessing plant stress, particularly in relation to temperature fluctuations [57,58]. High temperatures can accelerate chlorophyll degradation, which generally reduces photosynthetic efficiency [58,59]. However, under unfavorable conditions, many plant species reduce chlorophyll content to mitigate the harmful effects of high temperatures, which helps maintain stomatal conductance and consequently supports photosynthetic activity and growth [60].

Environmental manipulation studies are a widely applied approach for assessing the potential biological impacts of future climate change. In this study, such an experiment was conducted in an experimentally controlled growth room, where environmental conditions such as temperature were precisely regulated to simulate and evaluate plants’ responses to predicted climate scenarios. Unlike *in situ* experiments, these controlled environments provide consistent and reproducible conditions, enabling the precise regulation of variables to isolate specific plant responses.

This study aimed to investigate and compare the plastic responses of functional (SLA, LDMC) and mechanistic (SLWC, LT, SD, and chlorophyll content estimated with the RGB index I_1_) leaf traits expressed by the same clonal genotypes of *Iris pumila*, which were exposed to optimal and elevated (by 1 °C) air temperatures in a controlled growth environment. Specifically, the following questions were addressed: (1) What is the general pattern of plasticity in response to temperature increases? (2) Do functional and mechanistic leaf traits differ in the magnitude and direction of their plastic responses? (3) Is the correlation pattern between traits affected by temperature conditions? (4) Which category of traits, functional or mechanistic, is more sensitive to temperature change? By responding to these questions, the study intended to provide insights into the adaptive strategies of plants under thermal stress.

## 2. Results

### 2.1. Phenotypic Responses of SLA, LDMC, SLWC, LT, SD, and I_1_ to Temperature

To assess the phenotypic responses of leaf functional and mechanistic traits to temperature increases, the values of SLA, LDMC, SLWC, LT, SD, and I_1_ were determined in the same genotypes of *I. pumila* under the two temperature growth conditions—ambient and elevated by 1 °C. The mean values with the corresponding standard errors and coefficients of variation for the analyzed leaf traits are presented in Table 1. As can be seen, the phenotypic values of all analyzed leaf traits changed significantly with temperature. Regarding leaf functional traits, the mean SLA value was higher (up to 9%) and the LDMC value was lower (about 3%) in the leaves developed under the experimentally elevated temperature compared to the leaves of the same genotype developed at the ambient temperature. As for the leaf mechanistic traits, SLWC and LT decreased with the increase in temperature (both by 6%), while SD and I_1_ increased (up to 23% and even 67%, respectively).

### 2.2. Phenotypic Plasticity of SLA, LDMC, SLWC, LT, SD, and I_1_ to Temperature

The reaction norm plots for the SLA, LDMC, SLWC, LT, SD, and I_1_ of the *I. pumila* genotypes grown at ambient and elevated temperatures are depicted in Figure 1. As can be seen, the mean reaction norms are relatively steep, indicating the capability of *I. pumila* to adjust its leaf morphology and physiology to suit the prevailing temperature conditions. The pattern of reaction norms proved to be trait-specific. Thus, the reaction norms for SLA, SD, and I_1_ exhibited an upward trend with the increase in temperature, in contrast to those for LDMC, SLWC, and LT, which showed a downward trend (Figure 1). The reaction norms of individual genotypes exhibited a complex pattern, with some genotypes intersecting with each other, suggesting that besides their phenotypic values, their rank also changed at different temperatures. Additionally, a convergence trend was observed for the LDMC and I_1_ reaction norms. The reaction norms for LDMC converged towards a single value at the elevated temperature, in contrast to the ambient temperature, where their phenotypic values appeared to diverge. Conversely, the reaction norms for I_1_ converged towards a single value at the ambient temperature and diverged at the elevated temperature (Figure 1).

The individual variation of *I. pumila* genotypes, expressed as a coefficient of variation (CV%), changed in a trait- and temperature-specific manner (Table 1). Overall, the functional leaf traits showed less variation than the mechanistic traits. In addition, the functional traits exhibited a higher CV% at the ambient temperature, whereas the mechanistic traits exhibited a higher CV% at the elevated temperature. The highest CV% was observed for I_1_, while the lowest was found for LDMC. For example, the CV% for I_1_ was 44.3% and 47.1% at the ambient and elevated temperatures, respectively. Conversely, the CV% for LDMC was notably lower, amounting to 7.8% and 4.9% at the ambient and elevated temperatures, respectively. The CV% for SLA, LDMC, and LT was greater at the ambient temperature than at the elevated temperature, in contrast to the CV% for SLWC, SD, and I_1_. The results of an *F*-test for equality of variance confirmed that the CV% for LDMC and I_1_ differed between the alternative temperatures. However, these differences were only highly significant for I_1_ (*F* = 0.32; *p* < 0.001), but marginal for LDMC (*F* = 1.62; *p* = 0.058).

The mean phenotypic plasticity of SLA, LDMC, SLWC, LT, SD, and I_1_ under different temperature conditions was found to be trait-dependent (Figure 2). In general, the functional traits proved to be less plastic than the mechanistic ones (*PIv* = 0.239 vs. *PIv* = 0.369). Interestingly, the estimated plasticity index for the ratio of functional to mechanistic traits (*PIv* = 0.463) was found to be higher than the values considered for these traits individually. The most plastic traits were I_1_ and SD (*PIv* = 0.358 and *PIv* = 0.183, respectively), whereas the least plastic one was LDMC (*PIv* = 0.066). An intermediate level of plasticity was detected for SLA, SLWC, and LT (*PIv* = 0.104, *PIv* = 0.087, and *PIv* = 0.085, respectively). The results of a Friedman ANOVA showed that the amount of plasticity differed for all leaf traits analyzed *χ*^2^ (5, *N* = 240) = 67.11, *p* < 0.0001 (Figure 2).

### 2.3. Phenotypic Correlations Between Functional and Mechanistic Leaf Traits in Response to Temperature

The association between the phenotypic values of individuals for a pair of leaf traits is shown in Figure 3. Within each temperature treatment, a substantial number of significant correlations was observed, accounting for more than half of the total correlations. The positive correlation between SLWC and LT was the strongest among all correlations between pairs of leaf traits (r = 0.976 and r = 0.986, both *p* < 0.0001, in the ambient and elevated temperatures, respectively), whereas that between SLA and I_1_ was moderate in strength (r = 0.525, *p* < 0.001 and r = 0.546, *p* < 0.001 in the ambient and elevated temperatures, respectively). The correlation between SLA and SLWC was negative in sign and moderate in strength (r = −0.604 and r = −0.568, both *p* < 0.0001, in the ambient and elevated temperatures, respectively), as was the correlation between SLA and LT (r = −0.760 and r = −0.667, both *p* < 0.0001, in the ambient and elevated temperatures, respectively). The weakest negative correlation was detected between LDMC and I_1_ (r = −0.322, *p* = 0.04 and r = −0.353, *p* = 0.02 in the ambient and elevated temperatures, respectively).

Although the same total number of significant phenotypic correlations (two positive and six negative) was observed in both temperature treatments, the correlation structures suggest potential differences (Figure 3). For example, the significant correlation between SLA and LDMC (r = −0.663, *p* < 0.0001) as well as between SLWC and I_1_ (r = −0.345, *p* = 0.03) was only detected at the ambient temperature. Conversely, at the elevated temperature, significant correlations between SLA and SD (r = −0.318, *p* = 0.04) and between SD and I_1_ (r = −0.317, *p* = 0.04) were found. However, these differences do not appear to be strong enough for the correlation patterns between leaf traits expressed at ambient and elevated temperatures to differ significantly, as confirmed by the Mantel test (r = 0.902, *p* < 0.01).

Apart from the correlation analysis of the individual functional and mechanistic leaf traits, each trait category was also analyzed in its entirety. The first category, referred to as functional, showed a positive correlation between its phenotypic expression at ambient and elevated temperatures (r = 0.564, *p* < 0.001), as did the other category, referred to as mechanistic (r = 0.424, *p* < 0.01). In contrast, the functional and mechanistic trait categories were negatively correlated with each other during both temperature treatments (r = −0.794 and r = −0.819, both *p* < 0.0001, for ambient and elevated temperatures, respectively).

Furthermore, when correlating the plasticity indices, a strong and highly significant positive association was found between the SLWC and LT plasticity indices (r = 0.936, *p* < 0.0001). Additionally, a moderately positive association was discovered between the plasticity indices of SLA and I_1_ (r = 0.351, *p* = 0.03), as well as between those of LDMC and I_1_ (r = 0.338, *p* = 0.03). A marginally significant association was observed between the SLA and LT plasticity indices (r = 0.317, *p* = 0.05).

## 3. Discussion

### 3.1. Acclimation Responses to Increased Temperature

To withstand changing environments, plants have evolved diverse modifications in their morphology, physiology, phenology, and reproduction [61] or improved their capacity to tolerate and adjust to novel conditions through phenotypic plasticity [6,62]. The present study focuses on elucidating the role of leaf traits in response to elevated temperatures in *I. pumila*. By analyzing the phenotypic expression of its functional and mechanistic leaf traits at ambient and elevated temperatures, under controlled experimental warming conditions in a growth room, we aimed to assess the acclimation potential of *I. pumila* to adjust to new temperature environments. Despite *I. pumila* genotypes’ remarkable ability to respond plastically to variations in temperature conditions, both the magnitude and pattern of plastic responses appear to be highly trait-specific [63]. However, the observed trait specificity does not refer to the functional and mechanistic traits as comprehensive categories, but rather to individual traits within this categorization. This suggests that different leaf traits can exhibit different plastic responses regardless of their functional categorization, emphasizing the complexity of trait–environment interactions [63].

The two most important functional leaf traits, specific leaf area (SLA) and leaf dry matter content (LDMC), proved to be reliable indicators of plant resource utilization strategies in the sand dune environment [20,63,64,65,66]. Furthermore, SLA is generally positively associated with both leaf nitrogen content per unit of dry mass and assimilation rate, but negatively with leaf lifespan [67,68,69]. This indicates that the accumulation of dry matter is related to efficient photosynthetic activity, which is due to both transpiration fluxes and Rubisco activity [28,70,71]. Although there is substantial evidence that SLA increases with temperature [34,72], the relationship between SLA and temperature is not generally positive and can vary depending on plant species, ecosystems, and environmental conditions [73,74]. In the present study, the plants that were subjected to experimental warming generally exhibited greater SLA than the plants that were grown at the ambient temperature. This result is fully consistent with the findings of our previous study conducted *in situ* in natural populations of *I. pumila* [63]. Since only temperature was manipulated in the current experiment, it confirms that the change in SLA is a direct consequence of increased temperature [33]. Since increased temperatures can accelerate metabolic processes, such an increase in SLA could be due to increased resource allocation towards leaf area expansion, thereby optimizing photosynthetic efficiency under stressful conditions [66]. In addition, larger leaves enhance transpiration efficiency, which helps to cool the plant and maintain an optimal internal temperature [75]. Since gas exchange was not measured in this study, we can only assume that an increase in SLA serves as an acclimation mechanism to cope with heat stress by enabling more efficient transpiration and photosynthesis [76]. Future studies using direct measurements of stomatal conductance and photosynthetic rates would provide a more comprehensive understanding and further strengthen this hypothesis. LDMC reflects a plant’s investment in leaf structure, with higher values generally indicating tougher, more durable leaves that are resistant to environmental stresses [31,77,78]. A lower LDMC value is often associated with a strategy that favors resource acquisition and faster growth, as it corresponds to thinner, softer leaves that allow for faster carbon assimilation and water turnover. The relationship between LDMC and global warming is complex and context-dependent, implying that the response of LDMC may vary across plant species and ecosystems [36,37,38]. The decrease in LDMC at elevated temperatures observed in both this and our previous study [63] may indicate that plants are allocating fewer resources to the structural components of leaves, possibly favoring faster growth and increased metabolic activity [79]. In the context of the observed inverse relationship with SLA, this could indicate a trade-off between maximizing leaf area for photosynthesis (high SLA) and maintaining structural integrity or water retention (low LDMC), which could represent an adjustment mechanism to cope with higher temperatures.

Regarding the mechanistic leaf trait category, SLWC is closely responsive to global warming, as fluctuations in temperature and precipitation patterns affect water availability, which ultimately influences plants’ resilience to these challenges [40,80,81]. The observed reduction in SLWC may indicate various issues, such as increased transpiration, reduced water retention in leaf tissue, or a disturbed balance between water uptake and loss [63,82,83]. Overall, the decrease in SLWC with increasing temperatures can serve as a potential indicator of physiological stress and reflect the challenges the plant faces in coping with thermal stress. Conversely, the increased SD observed at elevated temperatures suggests that plants may adjust to these conditions by enhancing their gas exchange capacity and improving cooling through transpiration [84]. This response could serve as a compensatory mechanism to maintain photosynthesis under heat stress. Although SD provides a structural foundation for gas exchange, transpiration is primarily regulated by stomatal conductance and dynamic control of stomatal aperture, rather than density alone [50]. Studies have shown that stomatal conductance may increase with rising temperatures, thereby enhancing evaporative cooling and optimizing gas exchange efficiency [85]. A combined analysis of stomatal conductance and stomatal density would have provided a more comprehensive understanding of the physiological mechanisms underlying these responses, distinguishing structural adjustments from functional regulation in *I. pumila*’s acclimation to elevated temperatures. Given that even small changes in SD can significantly affect gas exchange and transpiration [86], the observed 20% increase with just a 1 °C rise in temperature suggests a remarkable acclimation response in *I.* pumila, improving water regulation. However, SD alone is insufficient for drawing definitive conclusions about transpiration, as stomatal function—particularly stomata’s ability to dynamically respond to environmental factors—plays a more critical role in regulating water loss. This underscores the need to consider both structural and functional aspects when studying plant transpiration mechanisms [50]. Leaf thickness (LT) is significantly affected by temperature, with higher temperatures generally leading to thinner leaves due to increased transpiration rates and reduced cell expansion [49,87]. The decrease in LT observed in this study probably reflects an acclimation strategy intended to protect the leaf tissue from overheating. Chlorophyll content, a key player in leaf photosynthesis, was significantly reduced under elevated temperature conditions, considering that the temperature increase was only one degree Celsius. This finding underlines the delicate balance that plants maintain with their environment and the potential implications of climate change on plant physiology. Even though some species do indeed reduce their chlorophyll content as a protective mechanism against heat stress [60,88], this study focused primarily on heat-induced damage to chlorophyll molecules and the photosynthetic machinery, which is a well-documented response in many species [89,90,91]. Unfortunately, no measurements of gas exchange were performed; these would have provided further insight into stomatal conductance and its role in photosynthetic activity under elevated temperatures.

### 3.2. Thermal Plasticity of Functional and Mechanistic Leaf Traits

Phenotypic plasticity has long been recognized as a key strategy that enables plants to adapt to variable environmental conditions [92]. As temperature is one of the most dynamic environmental factors and significantly affects plant metabolism, the plasticity of leaf functional traits is essential for maintaining physiological homeostasis and enhancing plant resilience to temperature fluctuations [93,94]. Our results confirm that the mean plastic responses of functional and mechanistic leaf traits, SLA, LDMC, SLWC, SD, I_1_, and LT, changed with the increase in temperature. Of the traits analyzed, the highest plasticity was observed in chlorophyll content (namely its proxy, I_1_), which suggests a strong capability of the plants to modulate their photosynthetic machinery in response to temperature fluctuations. The temperature sensitivity of photosynthesis is a crucial factor in predicting how plants will respond to warming [95]. By adjusting their photosynthetic apparatus, such as enzyme activity and membrane properties, to optimize carbon fixation under different environmental temperatures, plants will be able to maintain their productivity in changing climates [96].

Stomatal density (SD) was the second most plastic trait, reflecting the plant’s ability to optimize gas exchange under varying environmental conditions, such as humidity or CO_2_ concentration [55]. Modulating stomatal density helps the plant maintain a crucial balance between water loss and carbon dioxide uptake in response to fluctuating conditions. SLA, SLWC, and LT exhibited comparable levels of plasticity, which were relatively low in magnitude. This implies that these leaf traits remain relatively stable and show consistent expression across different thermal environments. The stability of these traits could be a result of physiological or ecological constraints, where maintaining consistent functionality is prioritized over flexibility in response to environmental changes [97]. In contrast, LDMC exhibited the lowest plasticity, suggesting that it represents a more conservative trait, being less responsive to short-term environmental changes and more reflective of a species’ long-term adaptation to its habitat. Low plasticity in LDMC is often associated with stress-tolerant species, as it reflects a resource allocation strategy that favors structural stability and resistance to abiotic stresses, such as drought [31]. The converging reaction norms discovered for LDMC at elevated temperatures further emphasize its limited plasticity.

In general, leaf traits related to light absorption and gas exchange, such as chlorophyll content and stomatal density, are more flexible and can rapidly adjust to environmental changes. In contrast, traits like LDMC, which reflect a plant’s resource use strategy and structural investment, are less plastic and may be more important for survival in stable or resource-limited environments.

In addition, our study provides evidence that the mean plastic response of mechanistic leaf traits, when considered collectively, is greater than that of functional leaf traits, indicating that mechanistic leaf traits have high sensitivity to temperature variation. This result is consistent with the perspective that mechanistic traits serve as physiological strategies that are specifically adjusted to the prevailing environmental factors and enable responses over short periods of time [98]. Higher plasticity allows plants to rapidly adjust to environmental changes, such as temperature fluctuations, enhancing survival in the short term. Conversely, functional leaf traits represent broader ecological strategies that respond to multiple environmental factors simultaneously and over longer periods of time [98]. Their relative stability and lower plasticity may result from their role in maintaining overall resource use efficiency and ecological balance across varying conditions. The observed higher plasticity of mechanistic traits emphasizes their role in fine-tuning physiological processes for acute stress management, whereas functional traits contribute to more generalized ecological strategies that prioritize consistency over flexibility. This dichotomy highlights the trade-off between rapid response and long-term stability in plant adaptation strategies [99,100,101].

### 3.3. Temperature-Driven Correlation Patterns of Leaf Traits

Since different traits often exhibit different plastic responses, it is likely that the correlations between them may also be shaped by environmental conditions [92,102]. Determining phenotypic correlations can provide important insights into the integration and stability of traits under varying environmental conditions [103]. It is well established that integrated phenotypes, that is, correlated suites of traits, can affect both ecological and evolutionary processes [104]. The high phenotypic correlations observed between leaf traits across different temperature conditions indicate a high degree of integration [105], suggesting that these traits function together in a coordinated manner. At the same time, the consistent phenotypic correlations across temperature treatments emphasize the stability of these leaf traits. Stable traits are likely to maintain their functional role despite environmental fluctuations and are, therefore, valuable targets for breeding resilient plant varieties [106,107].

To evaluate the integration and stability of leaf traits under varying temperature conditions, we assessed the phenotypic correlations between individual leaf trait expressions in each temperature treatment. The strongest positive correlation, observed between SLWC and LT at both temperature conditions, suggests a robust relationship between these two traits and may reflect an evolutionary adaptation in which thicker leaves improve water retention and overall structural integrity, contributing to higher leaf water content [108]. Moreover, the positive correlation of chlorophyll content with LDMC and LT, coupled with its negative correlation with SLA, suggests a trade-off in which thicker leaves invest more in chlorophyll to enhance photosynthetic efficiency, while thinner leaves prioritize maximizing light capture through their greater surface area [59]. In addition, we observed that different sets of leaf traits are correlated under distinct temperature conditions, highlighting the plants’ ecological plasticity and potential adaptability to different environmental pressures [109,110,111,112].

Apart from determining the correlation patterns of the individual functional and mechanistic leaf traits, the relationship between the two trait categories as a whole was also assessed. The strong inverse relationship between the mechanistic and functional trait categories observed at both temperature treatments suggests a trade-off in which a higher investment in one set of traits may come at the expense of the other. Some of the potential scenarios reflecting this balance between mechanistic and functional trait categories may include, for instance, photosynthetic efficiency vs. protection, growth vs. stress resistance, or efficiency vs. longevity [113].

It is believed that correlations between the plasticity of different traits arise when the given traits exhibit the same sensitivity to environmental conditions, share the same function, and/or have a genetic basis [92,102]. The existence of similar plastic responses allows for the maintenance of an integrated phenotype despite the influence of variable environmental conditions. Traits that exhibit significantly correlated plasticity indices appear to respond jointly to environmental changes, suggesting a coordinated adaptive response [114,115]. The positive correlation observed between the plasticity indices for SLA and LT, as well as between SLWC and LT, suggests that these traits respond jointly to environmental changes. The former implies that the plants may simultaneously adjust their leaf area and thickness to optimize light capture and structural support under varying environmental conditions. The latter may indicate that the plants employ coordinated adjustment mechanisms to optimize water storage and photosynthesis under changing conditions to ensure that they manage resources efficiently, for example by altering leaf structure or physiological processes. In this way, they can improve water use efficiency and maintain photosynthetic activity, ensuring survival and growth even under stressful conditions such as drought or high temperatures [116]. Conversely, the negative correlation between the plasticity indices of SLA and chlorophyll content, as well as between LDMC and chlorophyll content, indicates a trade-off in the plants’ adaptive strategies. If a plant adjusts its leaf area or dry matter content to optimize functions such as light capture or structural integrity, it may reduce its ability to modulate chlorophyll content and vice versa. This trade-off can help the plant to balance its resources and maintain its overall functionality under varying environmental conditions [117,118].

## 4. Materials and Methods

### 4.1. The Study Species

*Iris pumila* L. (Iridaceae) is a perennial, rhizomatous plant native to the Eurasian steppe with a wide geographical distribution. It extends from Austria in the west across central and southeastern Europe to the easternmost parts of western Siberia [119,120,121,122,123,124,125]. In Serbia, the species is native to the Deliblato Sands (44°90′23″ N, 21°11′32″ E), an isolated sand dune complex in the southern part of Banat [126], between the Danube and the western Carpathian slopes. Natural populations of *I. pumila* are widespread at exposed dune sites but can also be found in shaded habitats, usually under the canopy of *Pinus nigra*. Individual plants form rounded clones consisting of compact horizontally growing rhizome segments that extend from the center of a clone towards its margin. In such a way, the species establishes a rhizome system that can persist in an integrated state for many years [126]. In the Deliblato Sands, average annual temperatures have increased by 0.52 °C per decade, resulting in more arid conditions [127]. To reflect this trend, a 1 °C temperature increment was chosen in the present study to simulate conditions likely to occur in the near future.

### 4.2. Experimental Setup, Leaf Sampling, and Leaf Trait Measurement

The plants used for this study derived from a common garden experiment established back in 1986 from a natural population of *I. pumila* which inhabited a sun-exposed dune in the Deliblato Sands. In 2017, clonal genotypes from the experimental garden were individually planted into 500 cm^3^ plastic pots containing a 2:1 (*v*/*v*) mixture of soil substrate and sand. The potted plants were placed in randomized positions on shelves in a growth room with an automated temperature control system and grown under a 16 h photoperiod, with 110.5 µmolm^2^s^−1^ of photosynthetically active radiation (PAR) provided by Philips TLD 36-W/54 fluorescent lamps (Philips Lighting (Signify N.V.), Eindhoven, Netherlands). Plants were watered regularly, once a week, with a nutrient solution. The growth temperature was maintained at 23/19 °C (day/night) over multiple generations; this is, therefore, considered the ambient temperature. To minimize the effects of position, the pots were rotated twice a week.

In 2019, leaf samples from 40 randomly selected *I. pumila* genotypes were collected. The ambient temperature was then increased by 1 °C (24/20 °C day/night), and after approximately one month the newly formed leaves from the same genotypes were collected. Since the part of the leaf that is sampled (top, middle, or base) and its developmental stage influence the traits analyzed, we only sampled 2/3 of the upper part of the last fully developed leaf. At both sampling times, the experimental procedure included cutting off the last fully developed leaf from one ramet per genotype and immediately measuring its fresh mass. Afterwards, a digital image of the leaf area was captured by an optical scanner (Epson perfection V600, Epson, Suwa, Japan) and the projected leaf area was determined using the program ImageJ (v1.51j8) [128]. Subsequently, the leaf samples were oven-dried at 60 °C for 72 h to a constant mass and their dry mass was weighed.

The phenotypic values of the studied leaf traits were determined using standardized protocols for plant trait measurements [129]. SLA (in cm^2^ g^−1^) was estimated as the ratio of leaf area to leaf dry mass [77], LDMC (in g g^−1^) as the ratio of leaf dry mass to fresh mass [130], and SLWC as the difference between fresh and dry mass divided by leaf area [80]. LT was calculated according to the formula 1/SLA × LDMC [131].

Stomatal density (SD) was estimated using the impression method [132]. Briefly, an area in the middle section of the adaxial leaf surface was coated with a transparent nail polish. Once the polish was dry, it was peeled off with the adhesive tape and fixed on a microscope slide. The number of stomata was counted in 10 randomly selected microscopic fields (0.161 mm^2^ at 20 × 3.2 × magnification). *I. pumila* has amphistomatous, vertically oriented leaves, with no clear distinction between adaxial and abaxial leaf surfaces [133]. Since stomatal density does not differ between the two leaf surfaces (Appendix A), we arbitrarily selected the ‘adaxial’ surface—i.e., the side facing the rhizome—to ensure consistency across samples and to facilitate comparisons with our previous [53] and future studies.

The leaf chlorophyll content was estimated non-destructively using an image-based traits analysis [134]. Digital images of scanned leaves were analyzed for their green pixel intensity by extracting color information using an RGB model and ImageJ [128]. Previously, a pilot experiment (Appendix A) had shown that the RGB index I_1_ proposed by Sánchez-Sastre et al. [134] exhibits a moderately strong, but highly significant, linear negative correlation with the chlorophyll concentration, quantified by the DMSO method [135]. Furthermore, based on the regression analysis, which showed that all points are within the 99% prediction interval (Appendix A), it can be concluded that the non-destructively determined RGB index I_1_ provides reliable estimates of chlorophyll content, making it a valid alternative to the destructive method of chlorophyll quantification. Therefore, this particular index, I_1_, was used in the present study.

### 4.3. Statistical Analysis

All statistical analyses were performed with R Statistical Software (v4.2.3; R Core Team, 2023) in RStudio (v 2023.3.0.386) [136]: The *tidyr* [137], *dplyr* [138], and *reshape2* [139] packages were used to facilitate the data manipulation and reshaping. The *rstatix* package [140] was used to estimate the basic parameters of descriptive statistics (e.g., mean, standard error, coefficient of variation). The assumption of normality was checked and confirmed using the Shapiro–Wilk normality test from the base *stats* package [136]. Since each of the leaf trait measurements was completed on the same plant across two temperature regimes, a repeated measures ANOVA was performed using the *stats* package [136]. A comparison of the coefficients of variation (CV%) between the groups was performed using the *F*-test for equality of variances from the *stats* package [136]. The relationships between pairs of traits within a temperature regime were estimated using Pearson’s correlation analysis. The correlation coefficient matrices for leaf traits expressed at ambient and elevated temperatures, along with the corresponding correlation heatmaps, were created using the *corrplot* package [141]. Afterwards, the correlation matrices were compared using the Mantel test from the *vegan* package [142]. The temperature-induced plasticity in the analyzed leaf traits was determined by calculating the plasticity index, *PI*_V_ [143]:*PI*_V_ = |X_A_ − X_E_|/X_E_
where X_A_ is the value of a given leaf trait expressed at the ambient temperature, while X_E_ is the value of the same leaf trait expressed at an elevated temperature. *PI*_V_ denotes the measurement of the change in a trait from an elevated to the ambient temperature. A comparison of the plasticity indices of all analyzed leaf traits within the same plants grown under different temperature conditions was performed using Friedman’s ANOVA, as the data were not normally distributed. The significance of the difference between each two plasticity indices was assessed using the post hoc Wilcoxon rank-sum test from the *stats* package [136].

To perform statistical analyses on the individual trait categories—functional and mechanistic leaf traits—rather than on individual traits within each category, the data were first normalized due to the differing ranges of absolute values across the traits. Normalization is essential when integrating variables with different units or scales, as it ensures that each variable contributes equally to the analysis [144]. By normalizing the data, the influence of large-scale variables is reduced, allowing for more meaningful comparisons between traits [145]. Once normalized, the mean values for each trait category were calculated and used in correlation and plasticity analyses.

## 5. Conclusions

This study explores the role of phenotypic plasticity in enabling *Iris pumila* to adjust its functional and mechanistic leaf traits in elevated temperature conditions. By applying an experimental temperature manipulation design, our study confirmed that a temperature increase of 1 °C—a change comparable to that caused by global warming—is sufficient to induce a plastic response in functional and mechanistic leaf traits. The observed variation in the plastic responses of different leaf traits demonstrates the complex and trait-specific nature of plant acclimation strategies. Functional leaf traits, such as SLA and LDMC, and mechanistic traits, such as SD and chlorophyll content, exhibit different amounts of plasticity, reflecting the plant’s ability to adjust its resource utilization and physiological functions in response to thermal stress. Notably, mechanistic traits exhibit higher plasticity than functional traits, highlighting the rapid physiological adjustments that plants make in the face of immediate environmental stress. These results suggest that while plants are capable of responding to temperature changes, such adjustments may involve a trade-off between maximizing resource acquisition and maintaining structural integrity. Understanding the interplay between functional and mechanistic traits and their plastic responses provides valuable insights into plants’ resilience to climate warming, with implications for the prediction of ecological and evolutionary responses to future climate change.

## Figures and Tables

**Figure 1 plants-14-00960-f001:**
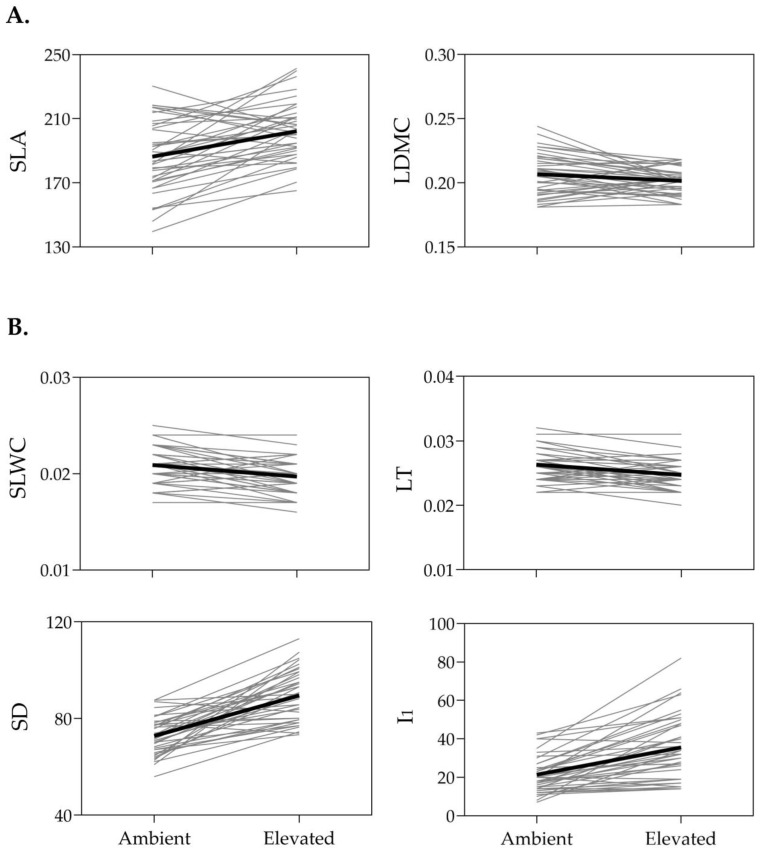
Reaction norm plots for the 40 *Iris* genotypes (fine lines) observed under two temperature treatments—ambient and elevated. The mean phenotypic plasticity is indicated by a heavy line. (**A**) Functional leaf traits: specific leaf area (SLA, in cm^2^ g^−1^), leaf dry matter content (LDMC, in g g^−1^). (**B**) Mechanistic leaf traits: specific leaf water content (SLWC, g cm^−2^), leaf thickness (LT), stomatal density (SD, no. stomata/mm^2^), and RGB index I_1_.

**Figure 2 plants-14-00960-f002:**
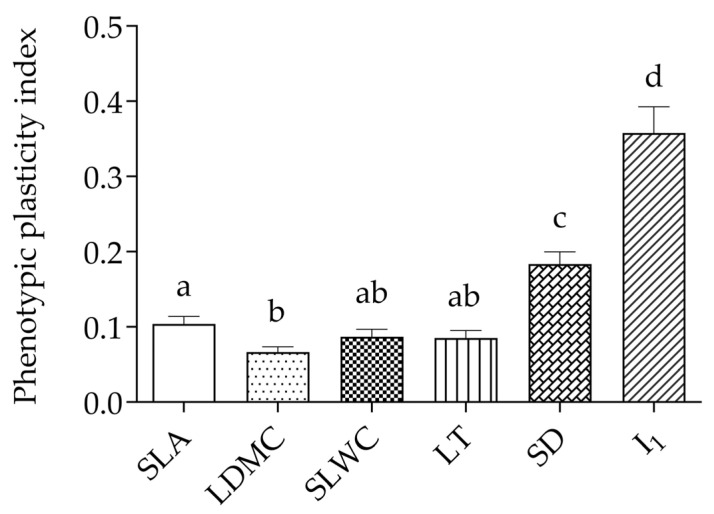
The phenotypic plasticity indices (mean ± SE) for the functional (SLA, LDMC) and mechanistic (SLWC, LT, SD, RGB index I_1_) leaf traits of 40 distinct *I. pumila* genotypes expressed at ambient and elevated air temperatures. Means with the same letter are not significantly different at *p* > 0.05, according to the Wilcoxon rank-sum test.

**Figure 3 plants-14-00960-f003:**
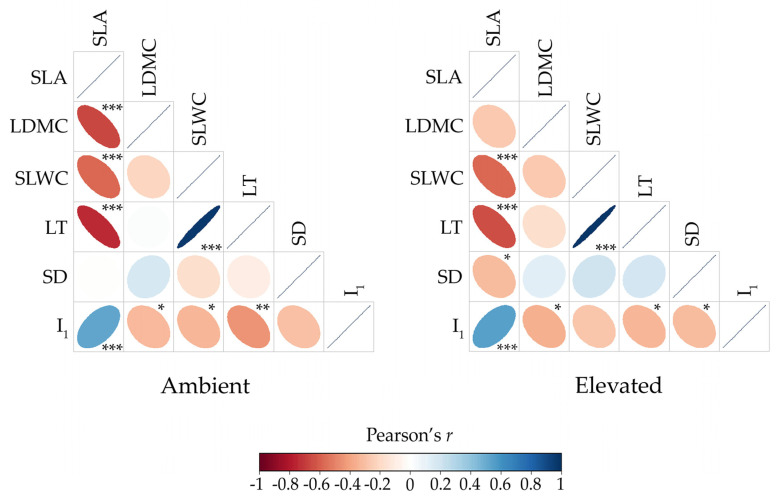
Heat maps of the correlations between the functional (SLA and LDMC) and mechanistic (SLWC, LT, SD, RGB index I_1_) leaf traits under ambient and elevated temperature conditions. The color gradient and ellipse eccentricity represent the strength and direction of the correlations, with blue indicating stronger positive correlations and red indicating stronger negative correlations. Correlation values are identified as weak (r < 0.4), moderate (0.4 ≤ r ≤ 0.8), or strong (r > 0.8). Asterisks indicate the significance level of the correlations: * *p* < 0.05, ** *p* < 0.01, *** *p* < 0.001.

**Table 1 plants-14-00960-t001:** The means ($\bar{X}$), standard errors (SE), and coefficients of variation (CV%) for the functional and mechanistic leaf traits: specific leaf area (SLA, in cm^2^ g^−1^), leaf dry matter content (LDMC, in g g^−1^), specific leaf water content (SLWC, in g cm^−2^), leaf thickness (LT), stomatal density (SD, no. stomata/mm^2^), and RGB index I_1_ of *Iris pumila* genotypes (*N* = 40) grown at ambient and elevated temperature (by 1 °C) conditions in an environmentally controlled growth room. The *F*-values of the temperature effects obtained using repeated ANOVA are presented for each trait as well: ns—non significant, * *p* < 0.05, **** *p* < 0.0001.

Leaf Trait		Ambient Temperature				Elevated Temperature				*F* for Comparison of Trait Means
		$$\bar{X}$$	SE	CV%		$$\bar{X}$$	SE	CV%		
* Functional *										
SLA		186.4	3.5	11.9		202.3	2.8	8.7		24.63 ****
LDMC		0.2066	0.0026	7.8		0.2014	0.0016	4.9		4.86 *
* Mechanistic *										
SLWC		0.0209	0.0003	9.2		0.0197	0.0003	9.4		18.95 ****
LT		0.0263	0.0004	9.2		0.0247	0.0004	8.9		25.37 ****
SD		72.8	1.2	10.5		89.5	1.6	11.6		83.89 ****
I_1_		21.3	1.5	44.3		35.5	2.6	47.1		45.14 ****

## Data Availability

The original contributions presented in this study are included in the article/Appendix A. Further inquiries can be directed to the corresponding author.

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
