# Peer review of "Plastic Responses of Iris pumila Functional and Mechanistic Leaf Traits to Experimental Warming"

_plants, 2025, doi:10.3390/plants14060960_

Round 1
Reviewer 1 Report
Comments and Suggestions for Authors
The aim of this work was to investigate the phenotypic plasticity of Iris pumila in response to the increase of 1°C under controlled conditions (110 umol/m2.s and 500 cm3/pot). Parameters evaluated were: specific leaf area (SLA), leaf dry matter content (LDMC), specific leaf water content (SLWC), stomatal density (SD), leaf thickness (LT) and chlorophyll content. It was found that the increased temperature induced specific plastic responses, with mechanistic traits exhibiting greater plasticity than functional traits, reflecting their role in short-term acclimation. SLA and SD increased at higher temperatures, while SLWC, LDMC, LT and chlorophyll content decreased under the warming conditions.
The manuscript is well-written and the results presented can be of interest to the readership of Plants. A point that drew the Reviewer’s attention was the experimental procedure. This species occur in sun-exposed areas, then, it seems that the PAR level used in the experiment was rather low.
Specifics:
Introduction:
=Chlorophyll content is a key indicator of plant productivity,… playing a central role in photosynthesis… and maintain stable photosynthetic activity =
The reviewer suggests reading the paper by Cutolo et al. 2023, as it seems that even a large reduction in Chl content should not, at least in theory, reduce electron transport (e.g. https://doi.org/10.1111/nph.19064).
Stomatal density:
= Environmental factors, especially temperature, play a crucial role in influencing stomatal density. For example, higher temperatures can lead to an increase in stomatal density as plants adjust to increase transpiration and cool their leaves=.
Regarding this topic, the reviewer suggests to consider the effect of vapor pressure difference, which reduces gs .
Material & Methods:
=In 2017, clonal genotypes from the experimental garden were individually planted into 500 cm3 plastic pots containing a 2:1 (v/v) mixture of soil substrate and sand. The potted plants were placed on shelves in the growth room at randomized positions and grown under a 16-h photoperiod and 110.5 μmol/m2.s of photosynthetically active radiation (PAR)…Plants were watered regularly, once a week, …. The growth temperature was maintained at 23/19°C (day/night), over generations, and is therefore considered the ambient temperature. …In 2019, leaf samples from 40 randomly selected I. pumila genotypes were collected. The ambient temperature was then increased by 1°C (24/20°C day/night), and after approximately one month the newly formed leaves from the same genotypes were collected. At both sampling times, the experimental procedure included cutting off the last fully developed leaf and immediately measuring its fresh mass=.
The pots used in the experiment were a little too small, which may negatively affect plant growth. Also, 1°C-gradient seems to be small for this kind of experiment. One can see, for example, the IPCC projections for this century.
However, the point that drew the reviewer ´s attention the most was the light conditions during the experiment. The light level used in the experiment seems to be very low (110 μmol/ m2.s). In particular considering that natural populations of Iris pumila are abundant in sun-exposed areas (Tucic et al. 2009, Pol. J. Ecol, 57:767-771).
Results:
If available, the reviewer suggests to present biomass data.
Also, Figure 1: The reviewer suggests to present these data as box plots (ambient , eCO2, and p value) instead of line charts.
Discussion
Lines: 409-412:
=Chlorophyll content plays a crucial role in the photosynthetic efficiency of leaves and is influenced by factors such as the leaf age and environmental conditions [83,84]. As temperatures increase, chlorophyll content decreases, resulting in reduced photosynthetic efficiency=
The reviewer suggests taking into account previous comments (see also the references mentioned earlier).
Conclusion:
It is suggested to observe the light conditions during the experiment.
Author Response
We sincerely appreciate your positive feedback regarding the quality of our manuscript, and we are encouraged that you recognized the potential interest of our results for the readership of Plants.
Comment 1: A point that drew the Reviewer’s attention was the experimental procedure. This species occur in sun-exposed areas, then, it seems that the PAR level used in the experiment was rather low.
Response 1: Thank you for your helpful criticism, as you pointed out which part needs more clarification. As far as light conditions are concerned, I. pumila inhabits mostly open steppe habitats in nature, but can also be found in shaded habitats where the average light intensity is between 45 and 135 µmol m-2s-1 (Vuleta et al 2010 https://doi.org/10.1093/jpe/rtp019). As this is a dwarf iris species whose leaves do not grow more than 10-20 cm high, even in open habitat it is usually overgrown by other plant species. Moreover, these plants have been successfully cultivated under these light conditions over the past several decades and the numerous experiments have been conducted by our research group (Avramov et al 2007 https://doi.org/10.1007/s11258-006-9207-3; Tucić et al. 2003, https://doi.org/10.1007/s10144-003-0137-9; Pemac and Avramov 2001 https://radar.ibiss.bg.ac.rs/handle/123456789/74; Pemac and Tucic 1998 https://doi.org/10.1007/BF00985227). For technical reasons, it is currently not possible for us to grow plants under different light conditions. Nevertheless, we believe that this intensity is appropriate for this type of experiment, as our aim was to determine the plasticity of the response of I. pumila leaf traits to changing temperature conditions without the influence of other environmental factors.
Comment 2: Introduction: =Chlorophyll content is a key indicator of plant productivity,… playing a central role in photosynthesis… and maintain stable photosynthetic activity =
The reviewer suggests reading the paper by Cutolo et al. 2023, as it seems that even a large reduction in Chl content should not, at least in theory, reduce electron transport (e.g. https://doi.org/10.1111/nph.19064).
Response 2: We are grateful for your insightful suggestion and for drawing attention to the paper by Cutolo et al. (2023). In the revised manuscript, we have updated the Introduction to include this perspective.
Comment 3: Stomatal density:
= Environmental factors, especially temperature, play a crucial role in influencing stomatal density. For example, higher temperatures can lead to an increase in stomatal density as plants adjust to increase transpiration and cool their leaves=.
Regarding this topic, the reviewer suggests to consider the effect of vapor pressure difference, which reduces gs.
Response 3: Thank you for highlighting the importance of vapor pressure difference (VPD) in influencing stomatal conductance (gs). We fully acknowledge that VPD is a critical environmental factor that can modulate stomatal behavior, particularly in response to temperature changes. Unfortunately, due to equipment limitations, we were unable to directly measure VPD or stomatal conductance in this experiment, and as such, it was not addressed in this study. We appreciate the reviewer’s insight, which could be explored in future research.
Comment 4: Material & Methods:
=In 2017, clonal genotypes from the experimental garden were individually planted into 500 cm3 plastic pots containing a 2:1 (v/v) mixture of soil substrate and sand. The potted plants were placed on shelves in the growth room at randomized positions and grown under a 16-h photoperiod and 110.5 μmol/m2.s of photosynthetically active radiation (PAR)…Plants were watered regularly, once a week, …. The growth temperature was maintained at 23/19°C (day/night), over generations, and is therefore considered the ambient temperature. …In 2019, leaf samples from 40 randomly selected I. pumila genotypes were collected. The ambient temperature was then increased by 1°C (24/20°C day/night), and after approximately one month the newly formed leaves from the same genotypes were collected. At both sampling times, the experimental procedure included cutting off the last fully developed leaf and immediately measuring its fresh mass=.
The pots used in the experiment were a little too small, which may negatively affect plant growth. Also, 1°C-gradient seems to be small for this kind of experiment. One can see, for example, the IPCC projections for this century.
However, the point that drew the reviewer ´s attention the most was the light conditions during the experiment. The light level used in the experiment seems to be very low (110 μmol/ m2.s). In particular considering that natural populations of Iris pumila are abundant in sun-exposed areas (Tucic et al. 2009, Pol. J. Ecol, 57:767-771).
Response 4: Thank you for your concern regarding the experimental procedure. I. pumila is a dwarf iris species whose leaves reach a size of 10 to 20 cm. Although in nature a clone can consist of a large number of ramets, the individual ramets are small and are grown for years in the growth room in 500 cm3 pots (Avramov et al 2007 https://doi.org/10.1007/s11258-006-9207-3 ; Tucić et al. 2003, https://doi.org/10.1007/s10144-003-0137-9 ; Pemac and Avramov 2001 https://radar.ibiss.bg.ac.rs/handle/123456789/74 ; Pemac and Tucic 1998 https://doi.org/10.1007/BF00985227 ). For this particular experiment, the plants have been successfully grown under these conditions since 2017. We would like to clarify that pots used in our experiment are not as small as one might assume based solely on the information about their volume. Their overall dimensions (i.e. their height and width) are perfectly suited to the growth of this species given the small size of I. pumila plants. The pots used were cylindrical, measuring 8 cm in diameter, 10 cm in height, and with a top surface area of approximately 50.27 cm². Each pot contained, on average, three ramets, with an average rhizome length of 2.5 cm. The total area occupied by the ramets was around 7.5 cm², leaving ample space for root and aboveground growth. We ensured that the pot size adequately supported both root development and the plant’s morphological structure. Based on these considerations, we are confident that the pot size did not negatively affected plant growth during the experiment.
In response to the comment regarding the temperature gradient appearing small for this type of experiment, we understand your concerns, especially when considering future climate projections. The gradual nature of temperature rise makes it particularly important to study how plants respond to small, incremental changes in temperature. Despite the modest change in temperature, this study aimed to investigate the plant's ability to perceive and respond to such changes, which our results clearly demonstrate. Furthermore, there is a lack of studies on the phenotypic plasticity of plants in response to minor temperature shifts, and this study provides valuable insight into how plants may cope with small but significant environmental changes (Jagadish et al, 2021 https://doi.org/10.1046/j.1420-9101.1998.11030285.x). Finally, we would like to emphasize that according to the 2019 IPCC report, which was relevant at the time our experiment was established, the average global mean surface temperature has increased by 0.87°C above pre-industrial levels. This increase justified our focus on investigating phenotypic responses to such a change. In addition, we also conducted a separate experiment under two warming regimes to simulate both lower and upper projected warming scenarios anticipated within the current century (Hocevar et al. 2023 https://hdl.handle.net/21.15107/rcub_ibiss_5892 ).
With respect to the comment about the light conditions during our experiment we acknowledge that the light intensity in our growth room (110 µmol/m²s) differed from the levels naturally observed in sun-exposed areas of Iris pumila populations. However, this intensity represented the maximum achievable given the technical limitations of the artificial light system available. While we acknowledge that these conditions may not perfectly replicate the natural environment, they ensured consistent and controlled experimental conditions throughout the study. Moreover, natural light levels for I. pumila populations are highly variable. This species usually inhabits open steppe habitats in nature, but can also be found in shaded habitats where the light intensity is rather low. For example, populations growing under the canopy of Pinus nigra experienced mean values of 132, 45 and 69 µmol/m²s in spring, summer and autumn, respectively (Vuleta et al. 2010). This variability points out that even within natural settings, I. pumila populations experience a wide range of light conditions, including levels similar to those used in our experiment. Additionally, the chlorophyll content measured in plant leaves in the growth room in a pilot experiment (with some results presented in the supplementary material) was 20.32 µg/cm² for chlorophyll a, 12.54 µg/cm² for chlorophyll b, and 32.86 µg/cm² for total chlorophyll. These values are consistent with chlorophyll levels observed under natural conditions (Tucić et al 2009, Pol. J. Ecol, 57:767-771).
Comment 5: Results: If available, the reviewer suggests to present biomass data.
Response 5: We have the data for the fresh leaf mass, but since it depended on the size of the harvested leaf, we do not believe that it is relevant. A more relevant option would have been the data about the total biomass of the entire ramet. However, since we did not remove the entire ramet in order to preserve it or further leaf sampling at elevated temperatures, we do not have measurements of the total biomass.
Comment 6: Also, Figure 1: The reviewer suggests to present these data as box plots (ambient , eCO2, and p value) instead of line charts.
Response 6: Reaction norms are typically visualized using line graphs or interaction plots, which effectively illustrate how a phenotype responds to different environmental conditions across various genotypes (Schlichting, Carl D., and Massimo Pigliucci. Phenotypic evolution: a reaction norm perspective. 1998). In a reaction norm plot, the x-axis usually represents the environmental variable (e.g., temperature), while the y-axis shows the phenotypic trait of interest. Each line represents the response of a specific genotype, making it easy to observe the differences in phenotypic plasticity across conditions. Box plots, on the other hand, are not well-suited for displaying reaction norms because they summarize data distributions (such as median, quartiles, and outliers) rather than showing continuous changes in a phenotype across environmental gradients. While box plots are useful for comparing the central tendency and variability of a trait across different groups, they do not provide information on the directional changes of individual genotypes or the relationships between environment and phenotype over a gradient. Thus, line graphs are preferred for illustrating reaction norms, as they allow for a clearer visualization of how each genotype's phenotype varies in response to environmental changes.
Comment 7: Discussion
Lines: 409-412:
=Chlorophyll content plays a crucial role in the photosynthetic efficiency of leaves and is influenced by factors such as the leaf age and environmental conditions [83,84]. As temperatures increase, chlorophyll content decreases, resulting in reduced photosynthetic efficiency=
The reviewer suggests taking into account previous comments (see also the references mentioned earlier).
Response 7: To avoid further misunderstandings and to summarize the Discussion section according to your comment and the suggestions of other reviewers, we have rephrased the entire paragraph.
Comment 8: Conclusion: It is suggested to observe the light conditions during the experiment.
Response 8: As we clarified in Responses 1 and 4 I. pumila inhabits mostly open steppe habitats in nature, but can also be found in shaded habitats where the light intensity is between 45 and 135 µmol m-2 s-1 (Vuleta et al 2010 https://doi.org/10.1093/jpe/rtp019). Moreover, these plants have been successfully cultivated under these light conditions for many years (Avramov et al 2007 https://doi.org/10.1007/s11258-006-9207-3 Pemac and Avramov 2001 https://radar.ibiss.bg.ac.rs/handle/123456789/74; Pemac and Tucic https://doi.org/10.1007/BF00985227 ). For technical reasons, it is currently not possible for us to grow plants under different light conditions. Nevertheless, we believe that this intensity is appropriate for this type of experiment, as our aim was to determine the plasticity of the response of I. pumila leaf traits to changing temperature conditions without the influence of other environmental factors. The light conditions in our experiment were held constant and optimized to ensure uniform growth conditions across treatments, allowing us to isolate the effects of temperature.

Reviewer 2 Report
Comments and Suggestions for Authors
This manuscript presents an insightful study on the phenotypic plasticity of Iris pumila leaf traits under elevated temperature conditions. The authors employ a well-controlled experimental design and provide a comprehensive analysis of functional and mechanistic leaf traits, offering valuable insights into the adaptive strategies of I. pumila. In conclusion, the paper would be sufficient to merit publication in Plants, though a revision is recommended, which needs to include the following points.
(1) Further clarification on the sampling protocol (e.g., number of leaves sampled per genotype) would improve reproducibility.
(2) The rationale for selecting a 1°C increase in temperature as the experimental treatment should be elaborated. While it aligns with global warming scenarios, an explanation of its ecological relevance to I. pumila’s native habitat is needed.
(3) The study involves raising the temperature by only 1°C in the growth chamber. However, it remains unclear whether the experimental setup ensures a consistently stable temperature differential between the two conditions. A detailed explanation of the growth chamber setup and the methods used to regulate temperature is necessary to validate the experimental conditions.
(4) In this study, leaves were sampled from plants grown at 23/19°C and compared with leaves that developed under the elevated temperature of 24/20°C after the plants were exposed to the new conditions. This method introduces the possibility that differences in plant age or leaf developmental stages may have influenced the results. It would be prudent to interpret the findings while considering such potential confounding factors inherent to this experimental design.
(3) minor comments
line 21: LT and chlorophyll -> LT, and chlorophyll
line 41: altered environment -> altered environments
line 137, 155,156, 227: SLWC, LT, SD and I1 -> SLWC, LT, SD, and I1
line 413: reduces photosynthetic efficiency -> reduces the photosynthetic efficiency
Author Response
We sincerely thank you for your positive and encouraging comments on our manuscript. We are pleased that you found our study on the phenotypic plasticity of Iris pumila leaf traits informative and thank you for your appreciation of the experimental design and comprehensive analysis. We greatly appreciate your feedback and are grateful for your constructive suggestions that helped us to improve the manuscript. We have carefully considered all the points raised and believe that the revisions have improved the quality of our work.
Comment 1: Further clarification on the sampling protocol (e.g., number of leaves sampled per genotype) would improve reproducibility.
Response 1: We have accepted your recommendation and provided additional clarification on the sampling protocol, including details such as the number of leaves sampled per genotype, in the Materials and Methods section to enhance reproducibility.
To ensure that all sampled leaves were at the same stage of development, only the last fully developed leaf per genotype was sampled. The leaf is considered fully developed when it is the youngest leaf with a small, newly emerging leaf at its base. Consequently, only one leaf per 40 genotypes and per temperature treatment was sampled. In I. pumila, it takes about one month for a new leaf to fully develop. Therefore, after increasing the temperature, the first fully developed leaf of 40 genotypes was sampled after one month. At your suggestion, we included this explanation in the text.
Comment 2: The rationale for selecting a 1°C increase in temperature as the experimental treatment should be elaborated. While it aligns with global warming scenarios, an explanation of its ecological relevance to I. pumila’s native habitat is needed.
Response 2: Thank you for this valuable suggestion. We have accepted this recommendation and added additional explanation for ecological relevance of selected temperature increase in the revised version of manuscript Materials and Methods.
In our previous study on the plasticity of leaf traits of I. pumila (Manitašević Jovanović et al. 2022, https://doi.org/10.3390/plants12173114), we focused on the open top chamber design to experimentally increase the temperatures (~1.5 °C) of I. pumila clones in the natural habitat. We observed the response of leaf traits to warming in two natural populations across seasons and years, without the regulation of other environmental factors. This study thus represents an addition to the research and aims to further clarify the plastic response of leaf traits under controlled conditions. Our aim was to investigate the general pattern of plasticity of leaf traits in response to an increase in temperature and to determine which category of traits - functional or mechanistic - is more sensitive to temperature changes and whether the correlation pattern between traits is affected by temperature conditions. Therefore, we argue that the presented experimental setup is suitable to achieve these goals.
Comment 3: The study involves raising the temperature by only 1°C in the growth chamber. However, it remains unclear whether the experimental setup ensures a consistently stable temperature differential between the two conditions. A detailed explanation of the growth chamber setup and the methods used to regulate temperature is necessary to validate the experimental conditions.
Response 3: Thank you for pointing out which section in Material and Methods needs to be better explained. The experimental setup was carried out in a plant growth room at our Institute specifically designed for controlled environmental studies. That is not a commercially available growth chamber with the specifications stated by the manufacturer but a customized system that allows for precise regulation of the temperature. The growth room is equipped with an automated temperature control system maintained with an air conditioner constantly monitored with two temperature probes. Thus, the temperature difference between the two setups was carefully monitored and kept constant at 1°C throughout the experiment. In addition, the pots were rotated twice a week to minimize positional effects.
We expanded the description of the growth room and the methods used for temperature regulation in the "Materials and Methods" section of the revised manuscript.
Comment 4: In this study, leaves were sampled from plants grown at 23/19°C and compared with leaves that developed under the elevated temperature of 24/20°C after the plants were exposed to the new conditions. This method introduces the possibility that differences in plant age or leaf developmental stages may have influenced the results. It would be prudent to interpret the findings while considering such potential confounding factors inherent to this experimental design.
Response 4: Thank you for raising this point. The developmental stage of the leaf as well as the specific part sampled (top, middle or base) undoubtedly influence the results of analyzed traits. Therefore, we took special care to ensure consistency by sampling leaves at the same stage of development. In our experiment, at both sampling times, we harvested exclusively the last fully developed leaf and always collecting the upper 2/3 of each leaf. In our experiment, we consistently sampled the youngest fully developed leaves to ensure that leaf age was standardized across all treatments. This approach eliminates the potentially confounding effect of differences in plant or leaf age on our results. By focusing on leaves of the same developmental stage, we ensured that any observed differences in leaf traits were due to the temperature treatment and not to differences in leaf maturity. In this way, our experimental design reliably isolates the temperature effects and allows a more accurate interpretation of phenotypic responses. At your suggestion, in the revised manuscript we included this explanation in the text of the Material and Methods section.
Comment 5: minor comments
line 21: LT and chlorophyll -> LT, and chlorophyll
line 41: altered environment -> altered environments
line 137, 155,156, 227: SLWC, LT, SD and I1 -> SLWC, LT, SD, and I1
line 413: reduces photosynthetic efficiency -> reduces the photosynthetic efficiency
Response 5: We have made all the suggested corrections in the text.

Reviewer 3 Report
Comments and Suggestions for Authors
Comments to the Author
This study analyzed the plastic responses of Iris pumila functional and mechanistic leaf traits to experimental warming. The research holds significant implications for understanding how plants adapt to climate change and, mechanistic or functional leaf traits show more plasticity to thermal stress. The following issues require the author's attention; please revise the article based on the following comments.
1. Line 61-65: Consider removing this section. If objective of the study is plasticity of specific plant traits in response to thermal stress then general statement about leaf traits and ecosystem functioning seems unnecessary here.
2. Line 66: It would be better to start this paragraph from “Functional traits encompass……….”and delete the first line of this paragraph.
3. Line 71-115: The introduction provided extensive detail on each functional and mechanistic leaf traits, making the section quite lengthy. I recommend condensing it to a more concise length by summarizing these traits in one paragraph and focusing on the most relevant information about how previous studies have shown variation of these leaf traits in response to temperature. The introduction provided extensive detail on each functional and mechanistic leaf traits.
4. Line 116-122: It would be better to present the experimental condition information in materials and methods section.
5. Line 131-133: "This information does not contribute directly to the main focus of this section. Consider removing these lines to maintain clarity.
6. Line 381-420: Rather than discussing each leaf trait individually in separate paragraphs, it would be more efficient and clearer to summarize the key findings related to all leaf traits in previous and your study. This approach would provide a more concise and coherent presentation of the results and also will improve the flow and readability of the text. Moreover, consider deleting the first line of about each trait in each paragraph that’s unnecessary.
7. Line 452-456: It would be more effective to present this information in the conclusion section by summarizing the key findings.
8. Discussion section is also too lengthy. It would be better to summarize the “subsection 3.3. Temperature-driven correlation patterns of leaf traits” by proving the most relevant information.
Author Response
We greatly appreciate that the reviewer, in general, recognizes the significance of these findings and appears to share our perspective on their importance. We want to thank you for your constructive comments and suggestions. In the revised version, we have carefully considered all your comments and made extensive corrections accordingly.
Comment 1: Line 61-65: Consider removing this section. If objective of the study is plasticity of specific plant traits in response to thermal stress then general statement about leaf traits and ecosystem functioning seems unnecessary here.
Response 1: We appreciate your suggestion, however we consider it essential, as it provides the crucial context for the selection of leaf traits as the focus of our study. This section bridges the gap between the more general effects of climate change on ecosystems and our specific investigation of phenotypic plasticity of functional and mechanistic leaf traits. Removing this section could disrupt the logical flow of the introduction and weaken the transition to the traits analyzed. To address these concerns, we have revised the section.
Comment 2: Line 66: It would be better to start this paragraph from “Functional traits encompass……….”and delete the first line of this paragraph.
Response 2: Thank you for the suggestion. After considering your comment, we have come to the conclusion that this sentence should be removed.
Comment 3: Line 71-115: The introduction provided extensive detail on each functional and mechanistic leaf traits, making the section quite lengthy. I recommend condensing it to a more concise length by summarizing these traits in one paragraph and focusing on the most relevant information about how previous studies have shown variation of these leaf traits in response to temperature. The introduction provided extensive detail on each functional and mechanistic leaf traits.
Response 3: We agree with the suggestion and summarized the Introduction by removing redundant sentences. The revised version focuses on key information about how previous studies have shown variation in these traits as a function of temperature.
Comment 4: Line 116-122: It would be better to present the experimental condition information in materials and methods section.
Response 4: Though we appreciate the reviewer’s suggestion, we believe that this section should be retained as it provides the necessary context for the reason this type of experiment was conducted. Rather than explaining the methodology in detail, this section introduces the rationale for our approach and explains its relevance in the context of climate change research. The specific experimental procedure is described in detail in the Materials and Methods section, but this brief explanation in the introduction section helps the reader to understand the study and justify our choice of methodology.
Comment 5: Line 131-133: "This information does not contribute directly to the main focus of this section. Consider removing these lines to maintain clarity.
Response 5: We believe that this sentence is important as it helps to consolidate the aims of the study and emphasize its importance. While we are open to rephrasing, we feel that its deletion could weaken the logical flow and clarity of the research objectives, making it less clear how answering these questions contributes to understanding the adaptive strategies of plants under thermal stress.
Comment 6: Line 381-420: Rather than discussing each leaf trait individually in separate paragraphs, it would be more efficient and clearer to summarize the key findings related to all leaf traits in previous and your study. This approach would provide a more concise and coherent presentation of the results and also will improve the flow and readability of the text. Moreover, consider deleting the first line of about each trait in each paragraph that’s unnecessary.
Response 6: We agree with the reviewer’s suggestion and have revised the text to present a more concise and coherent summary of the key findings related to all leaf traits, rather than discussing each trait individually in separate paragraphs. These changes have improved the flow and readability of the text. We have also deleted suggested sentences to further improve clarity.
Comment 7: Line 452-456: It would be more effective to present this information in the conclusion section by summarizing the key findings.
Response 7: Thank you for your thoughtful suggestion. We understand the importance of summarizing the key findings in the conclusion section for clarity and impact. However, we believe that this information is more appropriately placed in its current position as it helps to round off the discussion in this section and underpins the interpretation of the data presented. To address potential concerns about the phrase "To conclude" which may imply formal conclusions, we propose rephrasing it to ensure a smoother integration with the preceding text.
Comment 8: Discussion section is also too lengthy. It would be better to summarize the “subsection 3.3. Temperature-driven correlation patterns of leaf traits” by proving the most relevant information.
Response 8: Thank you for presenting your point of view. By addressing your previous suggestions, we have already shortened the discussion by removing certain sentences and rephrasing parts of the text, also taking into account the suggestions of previous reviewers. In this way, we were able to streamline the section while retaining its essential content. We have also summarized subsection 3.3. by highlighting most important results.

Round 2
Reviewer 1 Report
Comments and Suggestions for Authors
Regarding this manuscript, my opinion continues the same as shown in my previous review report.
Author Response
Comment: Regarding this manuscript my opinion continues the same as shown in my previous review report.
Response: We have carefully analyzed your previous review and made further revisions to enhance the clarity and strength of our argumentation.
Specifically, in the revised version:
- We have more clearly differentiated between stomatal conductance and stomatal density, reducing the emphasis on stomatal density as the sole determinant of transpiration.
- We have made a clearer distinction between transpirational areas and stomatal functioning.
- We have further clarified the discussion on chlorophyll reduction, incorporating alternative adaptive mechanisms to elevated temperatures to provide a more comprehensive perspective on plant responses to stress conditions.
We believe these revisions have further improved the manuscript, addressing the concerns raised in your earlier review.
Reviewer 2 Report
Comments and Suggestions for Authors
The paper is adequately improved according to the comments.
Comments on the Quality of English LanguageI am not a native speaker, so I cannot make a definitive judgment, but there do not seem to be any major issues with the English in the manuscript.
Author Response
Comment: The paper is adequately improved according to the comments.
Response: Thank you for your positive feedback. We appreciate your time and effort in reviewing our manuscript and are pleased to hear that the revisions have adequately address your comments.
Reviewer 3 Report
Comments and Suggestions for Authors
Thank you for your revisions. The manuscript has significantly improved in terms of clarity, and the revisions have effectively addressed the concerns raised. I recommend it for acceptance without further revisions.
Author Response
Comment: Thank you for your revisions. The manuscript has significantly improved in terms of clarity, and the revisions have effectively addressed the concerns raised. I recommend it for acceptance without further revisions.
Response: We sincerely appreciate your positive feedback and your recommendation for acceptance. Thank you for your thoughtful review and valuable comments, which have helped us improve the clarity and overall quality of the manuscript.